# Microencapsulation of Lead-Halide Perovskites in an Oil-in-Fluorine Emulsion for Cell Imaging

**DOI:** 10.3390/nano13091540

**Published:** 2023-05-04

**Authors:** Jia-Xin Wang, Chang Liu, Hao Huang, Rui He, Shengyong Geng, Xue-Feng Yu

**Affiliations:** 1Shenzhen Institute of Advanced Technology, Chinese Academy of Sciences, Shenzhen 518055, China; jx.wang@siat.ac.cn (J.-X.W.);; 2University of Chinese Academy of Sciences, Beijing 100049, China; 3Hubei Three Gorges Laboratory, Yichang 443007, China

**Keywords:** lead-halide perovskites, microencapsulation, oil-in-fluorine emulsion, cell imaging

## Abstract

The superior optical properties of lead-halide perovskites (LHPs) inspired significant research in cell imaging applications; many encapsulating processes have improved perovskite stabilities with comparable biosafety. Herein, facile solvent evaporation encapsulation based on an oil-in-fluorine emulsion for aqueous-stable and extremely nontoxic LHP microcapsules is described. Perfluorooctane dispersed the emulsifier fluorocarbon surfactant to form a continuous fluorine phase, while LHPs and polymethylmethacrylate (PMMA) were dispersed in 1,2-dichloroethane, then emulsified in the fluorine phase to form an oil-in-fluorine emulsion. CsPbBr_3_ microcapsules with a dense PMMA shell that protect fragile CsPbBr_3_ from the external environment and inhibit lead ion release were obtained after solvent evaporation. The CsPbBr_3_ microcapsules not only retained 91% of fluorescence intensity after exposure to water for 30 d but also possess extremely low cytotoxicity for MCF-7 cells. After exposure to 2 mg/mL of CsPbBr_3_ microcapsules for 48 h, the cell viability remained >90%. The intracellular uptake of CsPbBr_3_ microcapsules indicates its potential use in cell imaging.

## 1. Introduction

Lead-halide perovskites (LHPs) have attracted significant interest in bioimaging applications due to their superb optical properties [1]. Compared with traditional fluorescent nanomaterials, LHPs possess narrow and symmetric photoluminescence (PL) emissions, tunable emission wavelengths, and ultrahigh photoluminescence quantum yields [2,3,4,5,6], which facilitate multiple simultaneous fluorescent probe resolutions and satisfy noise sensitivity and multicolored optical imaging requirements [7]. Despite these fascinating features, LHPs are inadequate for bioimaging uses. Due to the ionic crystal structure, prolonged exposure to external factors such as moisture, light, oxygen, and polar solvents degrades LHPs, resulting in severe PL quenching that prevents long-term utilization [8,9,10,11]. Furthermore, the leakage of lead compounds severely reduces cell viability, so utilizing LHPs in bioimaging requires increasing LHP stabilities and minimizes lead leakage [12,13].

Microencapsulation is a widely utilized and powerful technology used industrially to enhance the stabilities of sensitive ingredients [14,15]. Unfortunately, as conventional emulsion-based encapsulating methods generally involve waterborne systems [16], encapsulation of water-sensitive LHPs remains a challenge. Alternatively, several examples of nonaqueous oil-in-oil emulsions based on alcohols or other polar solvents have been reported [17,18,19], but they do not avoid degradation and PL quenching of LHPs. Nevertheless, several novel encapsulation methods are effective for bioimaging applications [9,20,21,22,23,24]. Zhang and co-workers used drop-casting to prepare high-stability and nontoxic CsPbX_3_@microhemispheres as multicolor luminance probes [9]. Wang et al. used spray-assisted methods to prepare CsPbBr_3_@PMMA nanospheres with superior water resistance [20]. Ryu and co-workers reported another CsPbBr_3_−SiO_2_ synthetic method using a co-synthesized and double-encapsulation process [23]. However, the complicated synthetic procedures and high equipment requirements make efficient production extremely difficult. Thus, a straightforward, general, and effective encapsulation technique to form LHP microcapsules is of significant value for stable and nontoxic bioimaging applications. It requires the development of a mild nonaqueous emulsion system for LHP microencapsulation and stable dispersion of LHPs in the dispersed phase.

As a nonpolar medium, perfluorinated solvents are non-destructive to fragile perovskites and incompatible with common organic solvents [25], making them promising candidates for a continuous phase solvent to prepare LHP microcapsules. Therefore, perfluorooctane (PFO) was selected as the continuous phase solvent for the microencapsulation system, and 1,2-dichloroethane (DCE) dispersed the LHPs. In this oil-in-fluorine (O/F) system, the commercial fluorocarbon surfactant 1H,1H,2H,2H-perfluorooctyltriethoxysilane (POTS) served as the emulsifier. LHP microcapsules were prepared by solvent evaporation. Because the fluorocarbon surfactant plays a crucial role in the interfacial tension between the oil phase and fluorine phase, microcapsules of smaller sizes were prepared by adjusting the emulsifier concentration [26]. Due to the dense microcapsule shells acting as a physical barrier, inner LHPs were protected from an unfavorable external environment [15,27], which significantly improved the stability of LHP microcapsules and avoided lead ion leakage. This ensured long-term bioimaging utilization with extremely high cell viability during incubation. We evaluated the cytotoxicity of LHP microcapsules against MCF-7 cells and found that even after exposure to an ultrahigh microcapsule concentration of 2 mg/mL for 48 h, the cells maintained >90% survival rates, which indicated good biosafety.

## 2. Materials and Methods

### 2.1. Materials

All reagents were used as received without further purification. CsBr (99.99%), PbBr_2_ (99.99%), PbI_2_ (99.99%), PbCl_2_ (99.99%), oleic acid (OA, 99%), oleyl amine (OAm, 98%), N,N-dimethylformamide (DMF, 99.8%), toluene (99.8%), 1,2-dichloroethane (DCE, 99.8%) were purchased from Sigma-Aldrich. Perfluorooctane (PFO, 98%) and 1H,1H,2H,2H-perfluorooctyltriethoxysilane (POTS, >97%) were purchased from Aladdin. Dulbecco’s Modified Eagle’s Medium (DMEM), fetal bovine serum (FBS), penicillin-streptomycin (Pen-Strep), phosphate-buffered saline (PBS), and Hoechst33342 were purchased from Gibco.

### 2.2. Preparation of LHP Nanocrystals

The LHPs were synthesized following a previously reported room temperature method [28,29]. CsPbBr_3_ nanocrystals were synthesized by supersaturated crystallization. The CsPbBr_3_ perovskite precursor solution was prepared by adding 73.40 mg CsBr, 42.55 mg PbBr_2_, 50 µL OLA, and 0.5 mL OA to 5 mL DMF and sonicating the mixture until it became transparent. A 1 mL aliquot of the precursor solution was rapidly injected into 10 mL of toluene with vigorous stirring. The CsPbBr_3_ colloidal solution was centrifuged for 10 min at 12,000 rpm to obtain CsPbBr_3_ precipitate to use in anion-exchange reactions and encapsulations.

The CsPbCl_x_Br_3−x_ and CsPbI_x_Br_3−x_ quantum dots were obtained by anion exchange. PbCl_2_ and PbI_2_ as anion sources were dissolved in DCE with OAm and OA. A total of 10 mg of CsPbBr_3_ dispersed in 5 mL DCE was used for anion exchange. To acquire CsPbCl_x_Br_3−x_, 1.1 mg PbCl_2_ was dissolved in 0.4 mL DCE with 40 µL OAm and 40 µL OA, then the mixture was added to 5 mL CsPbBr_3_ (2 mg/mL) DCE dispersion with stirring for 30 min. To acquire CsPbI_x_Br_3−x_, 4.6 mg PbI_2_ was dissolved in 1 mL DCE with 100 µL OAm and 100 µL OA, then the mixture was added to 5 mL CsPbBr_3_ (2 mg/mL) DCE dispersion with stirring for 5 min. After completion of the reaction, the CsPbCl_x_Br_3−x_ and CsPbI_x_Br_3−x_ perovskites were isolated by centrifugation. The obtained precipitates exhibited different emission colors.

### 2.3. Encapsulation of LHP Microcapsules

The LHPs were centrifuged and encapsulated by solvent evaporation as shown in Figure 1. In contrast with the traditional process [30], a perfluorinated solvent formed the continuous phase solution in this system. In general, POTS was mixed with various PFO emulsifier concentrations to form a fluorine-phase solution; the dispersed oil-phase solution was prepared by dispersing 0.1 g of the perovskites and 0.25 g of polymethylmethacrylate (PMMA) in 5 mL DCE. Then, 1.5 mL of the dispersed phase solution was added to 10 mL of the fluorine phase solution and stirred at 10,000 rpm for 3 min to form a stable oil-in-fluorine emulsion. After evaporating the dispersed phase solution of DCE by using rotary evaporation method, a dense PMMA shell formed at the oil–fluorine interface, which isolated the fragile perovskite core material from the environment. The LHP microcapsules were separated by a vacuum filtering process using a filter membrane (pore size, 0.1 µm).

### 2.4. Material Characterization

The particle size distribution of the microcapsules was measured using a Zetasizer Nano ZS90 laser particle size analyzer at room temperature. The morphologies of the CsPbBr_3_ microcapsules were characterized using a Zeiss Sigma 300 field emission scanning electron microscope (FE-SEM). The lead concentration in deionized water was measured by an inductively coupled plasma mass spectrometry (ICP-MS) (using Octopole reaction system, Agilent model 7500ce). Fourier-transform infrared spectra (FT-IR) were collected in the transmittance mode on a Nicolet IS10 FTIR spectrophotometer. X-ray diffraction (XRD) data were collected on a Bruker D8 Advance diffractometer. UV-Vis absorption spectra were recorded on a Shimadzu Corporation UV-3600 spectrophotometer. Photoluminescence (PL) spectra and sample PL decay curves were excited using a 405 nm LED fluorescence spectrophotometer (Edinburgh FSL1000).

### 2.5. Cell Viability Assay and Cell Imaging

MCF-7 cells were seeded in a 48-well plate with 1 × 10^4^ cells per well and cultured in DMEM, supplemented with 10% FBS and 1% Pen-Strep in a 5% CO_2_ humidity incubator at 37 °C. CsPbBr_3_ microcapsules were ultrasonically dispersed in a PBS solution. After removing the medium, MCF-7 cells were incubated in the presence of CsPbBr_3_ microcapsules at concentrations from 0 to 2 mg/mL for 24 or 48 h. For the cell viability test, a 10% Cell Counting Kit-8 (CCK-8) was added to each well, and the cells were incubated at 37 °C for 2 h. After incubation, the absorbance at 450 nm was detected by an enzyme marker (FilterMax F5, Molecular Devices).

For cell imaging, the MCF-7 cells were seeded on confocal dishes (5 × 10^4^ cells) overnight and then incubated in CsPbBr_3_ microcapsules-containing medium at 0.25 mg/mL for various durations (6 and 12 h). The cells were washed with a PBS solution and stained with Hoechst33342 for 10 min before confocal fluorescence imaging.

## 3. Results and Discussion

Similar to hydrocarbon surfactants [31], fluorocarbon surfactants are important factors that affect microcapsule preparations. Amphiphilic fluorocarbon surfactants with fluorophilic and lipophilic groups dissolve both phases of an O/F emulsion system well via intermolecular interactions due to different polarities in the fluorine and oil phases [32]. By adjusting the fluorocarbon surfactant concentration, the tension at the fluorine and oil phase interfaces changes, and the emulsion stability changes as a result. To investigate the relationship between fluorocarbon surfactant stability on the emulsion system and the particle size distribution of the microcapsules, microcapsules were prepared using a series of fluorine phase solutions with POTS volume concentrations from 5 to 30%. Figure 2a shows the droplet stability. After settling for 5 min, the 5% POTS volume emulsion exhibited a distinct phase separation, while the phase separation degree dropped in the 10% POTS emulsion. Increasing the emulsifier volume concentrations to 20 and 30% resulted in no emulsion phase separation within 5 min. Consequently, increasing the emulsifier concentrations may result in a more stable O/F emulsion.

The particle size distribution of the microcapsules was measured by dynamic light scattering. As shown in Figure 2b, two peak regions were observed in the 5% POTS concentration, while the broadest and the narrowest particle size distributions were observed at POTS concentrations of 10% and 30%, respectively.

Figure 2c shows the z-average diameter of the CsPbBr_3_ microcapsule particle sizes; the mean particle size shrunk with increasing POTS concentrations. The z-average diameters of particle sizes are 480, 400, 290, and 210 nm at POTS concentrations of 5%, 10%, 20%, and 30%, respectively.

The SEM image in Figure 2d shows a representative morphology of the CsPbBr_3_ microcapsules. The relatively large microcapsule particle size was ~200 nm with a smooth and clean surface. Moreover, the surfaces of the smaller microcapsule particles were equally smooth and clean, which suggested that CsPbBr_3_ perovskites were embedded in the core. Furthermore, 5 mg of CsPbBr_3_ microcapsules were immersed in 5 mL of deionized water for 6 h and 12 h, and the lead concentrations in the deionized water were measured by ICP-MS. As shown in Appendix A, the lead concentration is extremely low after 6 h (3.1 ppb), suggesting a trace amount of CsPbBr_3_ residual on the surface. When the immersing time was prolonged to 12 h, the final lead concentration slightly increased to 4.1 ppb, proving that CsPbBr_3_ microcapsules with the PMMA shell effectively protect Pb^2+^ ions from dissolution in water.

In addition, Figure 2e shows the FT-IR spectra of CsPbBr_3_ microcapsule powder, PMMA, and POTS. For CsPbBr_3_ microcapsules, the peaks at 2950, 1722, 1434, and 1142 cm^−1^ come from PMMA and correspond to CH stretching, C=O stretching, CH_3_ stretching, and —O—CH_3_ stretching vibrations, respectively [33,34]. For POTS, peaks at 1144, 1238, and 1392 cm^−1^ bands were rocking vibration peaks of C—F bonds. The peak appearance at 1081 cm^−1^ was assigned to Si—O—Si bonds [35]. No characteristic absorption peaks of POTS were observed in the FT-IR spectra of LHP microcapsules, which indicated no fluorocarbon surfactant encapsulating in microcapsules.

Powder X-ray diffraction (XRD) patterns of CsPbBr_3_ microcapsules, pure CsPbBr_3_ perovskite, PbBr_2_, and CsBr are presented in Figure 2f. A series of strong diffraction peaks at 21.42, 28.61, 30.67, 34.37, 34.7, 43.68, and 46.52° are observed, assigned to the (121), (221), (202), (114), (321), (242), and (411) lattice planes, respectively [36]. Despite this, PbBr_2_ and CsBr were not detected in the CsPbBr_3_ microcapsules, which confirmed that encapsulation caused no damage to the perovskites. The XRD patterns of CsPbCl_x_Br_3−x_ and CsPbI_x_Br_3−x_ microcapsules are presented in Appendix A. When the halide ions were changed from Cl^−^ to Br^−^ and to I^−^, the diffraction peaks shifted toward a smaller angle direction, which phenomenon is in accordance with the previous studies [5,37].

Microcapsules with different emission colors were successfully prepared by adjusting the halide ratios in the LHP nanocrystals. Appendix A shows the photograph and PL spectra of these microcapsules. The luminescence properties of LHP microcapsules were comprehensively studied. Figure 3a shows the UV-Vis absorption spectra for both pristine CsPbBr_3_ and CsPbBr_3_ microcapsules exhibited a broad range of intense absorptions. The CsPbBr_3_ microcapsule emission peak occurred at 524 nm, pristine CsPbBr_3_ at 527 nm, and both had full width at half maximum (FWHM) ~19 nm. As shown in Figure 3b, the double-exponential characteristic was observed in the time-resolved PL spectra of the CsPbBr_3_ microcapsules and pristine CsPbBr_3_. The biexponential fitting parameters (with a model of y = A_1_ exp(-x/τ_1_) + A_2_ exp(-x/τ_2_)) [36] are summarized in Appendix A. The average PL lifetime of CsPbBr_3_ microcapsules was 15.98 ns, while that of pristine CsPbBr_3_ was 9.61 ns. Surface quenching often leads to shorter lifetimes [38,39]. The longer lifetime implies that there are fewer surface traps in the CsPbBr_3_ microcapsules.

The stability of LHP microcapsules was investigated. Figure 3c and Appendix A show that the PL intensity of the CsPbBr_3_ microcapsules remained ~91% after exposure to air for 30 d, while the pristine CsPbBr_3_ retained only 12% fluorescence intensity after 2.5 d. Furthermore, CsPbBr_3_ microcapsules were soaked in water for 30 d. Water rarely affected the PL intensity of the CsPbBr_3_ microcapsules, and the fluorescence intensity only declined by ~9%. However, owing to the ionic nature of CsPbBr_3_ crystals and high solubility of CsBr in water, CsPbBr_3_ exhibit an extremely low stability against water [40,41]. The fluorescence intensity of pristine CsPbBr_3_ rapidly declined by ~86% upon soaking in water for 20 min (Appendix A). Indeed, CsPbCl_x_Br_3−x_ and CsPbI_x_Br_3−x_ microcapsules exhibited similar stability for 30 d in air and water, while the intensity of fragile CsPbCl_x_Br_3−x_ and CsPbI_x_Br_3−x_ declined rapidly even after exposure to just 100 µL of water in the colloidal (Appendix A). Moreover, upon treatment with acid, alkali, or H_2_O_2_, PL emission of the CsPbBr_3_ microcapsules remained relatively stable for up to 15 days (Figure 3d). These results illustrate the extraordinary aqueous-media soaking stability of the LHP microcapsules, attributed to the protection of the dense PMMA shell.

The cell viability assay using a CCK-8 kit was performed on MCF-7 cells to assess the cytotoxicity of CsPbBr_3_ microcapsules for cell-imaging applications. After exposure to CsPbBr_3_ microcapsules at clinically relevant concentrations between 0 and 2 mg/mL for 24 h, cell viability remained ~100% (Figure 4a), which indicated that CsPbBr_3_ microcapsules were not cytotoxic. The incubation time was extended to 48 h to evaluate the long-term cell viability of CsPbBr_3_ microcapsules. As shown in Figure 4b, 98.3%, 97.4%, 95.4%, 92.1, and 90.5% cell survival rates were observed after CsPbBr_3_ microcapsule incubations at doses of 0.1, 0.25, 0.5, 1, and 2 mg/mL, respectively. These results confirmed that CsPbBr_3_ microcapsules exhibit low toxicity to cells and are suitable for cell imaging.

The fluorescence imaging studies investigated the application of CsPbBr_3_ microcapsules in the cell imaging using confocal laser scanning microscopy. MCF-7 cells were incubated with 0.25 mg/mL of CsPbBr_3_ microcapsules as fluorescent probes for different times. Afterwards, the nuclei of living cells were stained by Hoechst 33342, which was excited by a 346 nm laser to emit blue fluorescence. As shown in Figure 4c, after 6 h of incubation, the green fluorescence signals from CsPbBr_3_ microcapsules were observed upon being excited by a 405 nm laser. More importantly, the green fluorescence was distributed surrounding cell nucleus, indicating that CsPbBr_3_ microcapsules were aggregated in the cytoplasm as fluorescence probes. As larger microcapsules normally encapsulate more CsPbBr_3_ grains, the fluorescence of microcapsules with larger sizes may be brighter and cover the fluorescence of the microcapsules with smaller sizes during image acquisition. After extending the incubation time to 12 h, the green fluorescence was still strong, and the morphologies of the CsPbBr_3_-microcapsules-treated cells remained unchanged (Figure 4d). Based on these results, we concluded that CsPbBr_3_ microcapsules may work as a potential fluorescence probe for cell imaging applications with well biosafety.

## 4. Conclusions

Based on a nonaqueous oil-in-fluorine (O/F) emulsion system, we introduced an efficient solvent evaporation encapsulation method to prepare LHP microcapsules. Due to the solvent sensitivity of LHPs, nonpolar PFO served as the continuous phase, DCE as the dispersed phase, and the fluorocarbon surfactant POTS as the emulsifier for the O/F emulsion system. Notably, increasing the fluorocarbon surfactant concentration reduced the oil–fluorine phase interfacial tension. Consequently, the emulsion stability improved, and the microcapsule particle sizes decreased. Compared with the pristine CsPbBr_3_, CsPbBr_3_ microcapsules exhibit almost consistent luminescence properties. Furthermore, CsPbBr_3_ microcapsules have longer lifetimes, which indicates a decrease in the perovskite surface trap density. Due to the dense PMMA shell, the CsPbBr_3_ microcapsules retained 91% of fluorescence intensity after exposure to water for 30 days and possessed extremely low cytotoxicity on MCF-7 cells. After 6 h of coincubation, the CsPbBr_3_ microcapsules were efficiently endocytosed by cells, and strong fluorescence signals were observed. Therefore, CsPbBr_3_ microcapsules may work as a potential fluorescence probe for cell imaging applications.

## Figures and Tables

**Figure 1 nanomaterials-13-01540-f001:**
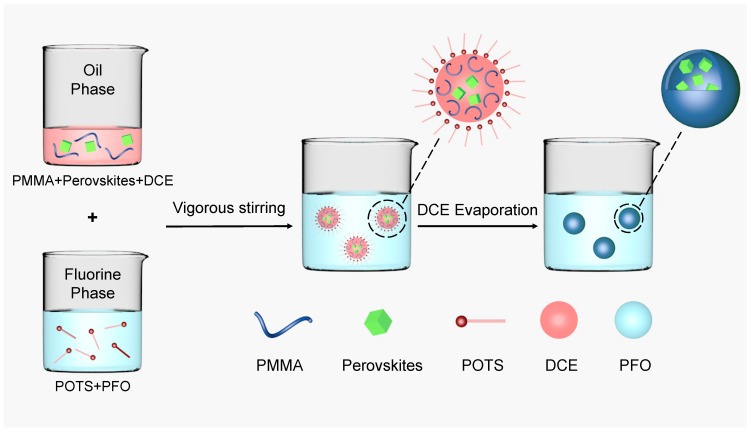
Illustration of the LHP microcapsule synthesis.

**Figure 2 nanomaterials-13-01540-f002:**
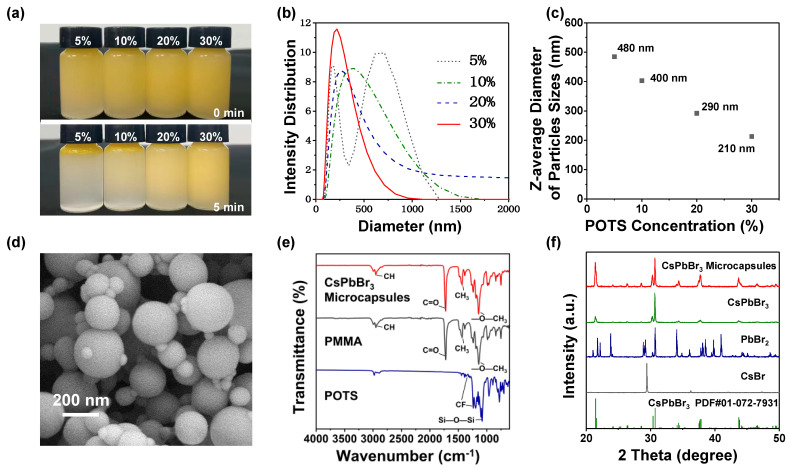
(**a**) Oil-in-fluorine emulsion prepared at different POTS concentrations. Photos were taken immediately (**top row**) and 5 min after initial emulsification (**bottom row**); (**b**) the effect of POTS concentrations on the particle size distribution of CsPbBr_3_ microcapsules; (**c**) the mean particle diameter; (**d**) scanning electron microscopy (SEM) image of the CsPbBr_3_ microcapsules; (**e**) FT-IR spectra of CsPbBr_3_ microcapsules, PMMA, and POTS; (**f**) XRD pattern for the CsPbBr_3_ microcapsules, pure CsPbBr_3_ perovskite, PbBr_2_, and CsBr.

**Figure 3 nanomaterials-13-01540-f003:**
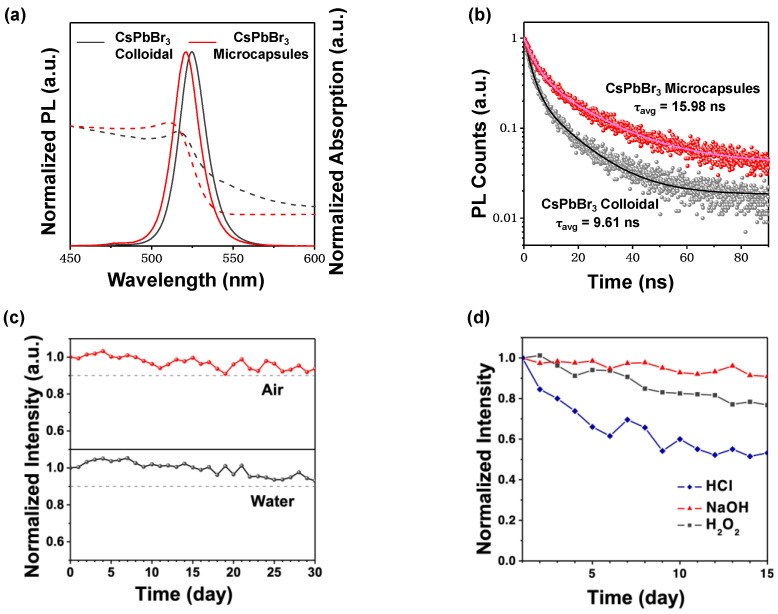
(**a**) UV–Vis absorption (dash lines) and PL emission (solid lines) spectra of pristine CsPbBr_3_ and CsPbBr_3_ microcapsules; (**b**) time-resolved PL decays of pristine CsPbBr_3_ and CsPbBr_3_ microcapsules; (**c**) relative fluorescence intensity of CsPbBr_3_ microcapsules as a function of time in air or water; (**d**) relative fluorescence intensity of CsPbBr_3_ microcapsules as a function of time in different aqueous-media solvents.

**Figure 4 nanomaterials-13-01540-f004:**
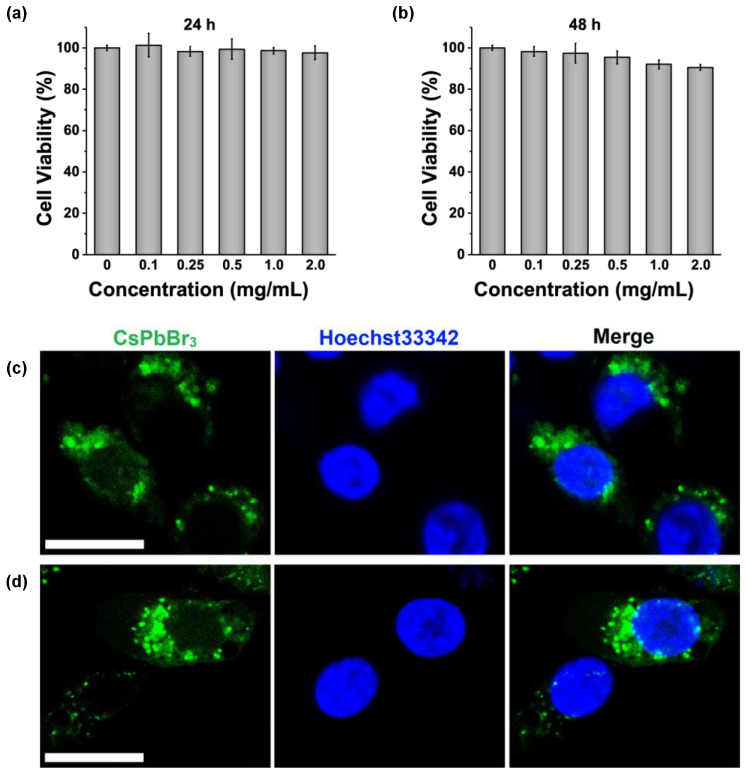
(**a**) Cellular viability of MCF-7 cells after 24 h and (**b**) 48 h of incubation with CsPbBr_3_ microcapsules; (**c**) confocal fluorescence microscopy images of MCF-7 cells after 6 h and (**d**) 12 h of incubation with CsPbBr_3_ microcapsules. Scale bars are 20 µm.

## Data Availability

The data presented in this study are available on request from the corresponding author.

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
