# Peer review of "Microencapsulation of Lead-Halide Perovskites in an Oil-in-Fluorine Emulsion for Cell Imaging"

_nanomaterials, 2023, doi:10.3390/nano13091540_

Round 1

Reviewer 1 Report

The manuscript of Jia-Xin Wang et al. brings a procedure for the stabilization of Cs-Pb-halide perovskite nanocrystals in liquid media. The particles are successfully used for cell labelling and negligible toxicity for cells is shown. The topic of the article is actual and the manuscript brings new data. The work is well designed and the paper is clearly and straightforwardly written. However, in my opinion, the manuscript lacks a detailed experimental procedure, which should be added before its publication.

Objections:

The procedure for drying the microcapsuled nanoparticles should be provided in more detail. Figure 1 is misleading – does “evaporation” mean that all solvents were evaporated, or only DCE was evaporated? (lines 107–109) How were the particles dispersed in water and in aq. media for cell labelling?

How was an exchange of halides proved? How much of LHP was dissolved (5-10 mg, should be specified). (line 93) Was the formation of LPH containing mixed halides tested starting from CsCl, PbBr2 and PbX2 mixture? The materials with mixed halides are not fully characterized and are not utilized – add full characteristics or avoid them.

What compound present in the suspension of pristine LHP quenches the luminescence? Water? Or, is the material partly dissolving? Please, clarify or add suggestions.

From Fig. 4c,d it is not clear that nanoparticles entered the cells; by eye, it seems that most of them stood on the cellular surface, which is controversy with the authors´ statements. (lines 235, 255)

Minor:

I recommend using 2 mg/mL rather than 2,000 .mu.g/mL to avoid confusion (decimal place). (lines 21, 71, 227)

Particle size is declared with non-reasonable accuracy (concentration of the starting mixture is given at 2 valid digits, but the size read from a broad peak distribution is given up to 4 digits. (line 156, Fig. 2c)

line 178: …caused no damage…

Reviewer 2 Report

The Manuscript entitled 'Microencapsulation of Lead-halide Perovskites in an Oil-in-fluorine Emulsion for Cell Imaging' describes the development of an efficient solvent evaporation encapsulation method for the preparation of LHP microcapsules. I can recommend the manuscript for publication if the authors make a few changes to the text of the manuscript.

1) The terminology "perovskite" is not entirely appropriate in the manuscript where it refers to a compound. For example, "After completion of the reaction, the perovskites were isolated by centrifugation." The material has the structure of a perovskite. A compound with perovskite structure (material) can be isolated by centrifugation.

2) «The elemental distribution on the surface of the CsPbBr3 microcapsules was characterized by an Energy Dispersive Spectrometer (EDS).»  EDX is not suitable as a surface survey technique, as X-rays are generated in an area around 2 microns in depth.  The size of the microcapsules is much smaller than this, Figure 2.

3) The main peak maps for PbBr2 and CsBr compounds in Figure 2f should be given to support the claim that these compounds are not present in CsPbBr3 microcapsules.

4) There is an error in the text, X-ray patterns are shown in Figure 2f.
